# COVID-19 Mortality among Hospitalized Patients: Survival, Associated Factors, and Spatial Distribution in a City in São Paulo, Brazil, 2020

**DOI:** 10.3390/ijerph21091211

**Published:** 2024-09-14

**Authors:** Marília Jesus Batista, Carolina Matteussi Lino, Carla Fabiana Tenani, Adriano Pires Barbosa, Maria do Rosário Dias de Oliveira Latorre, Evaldo Marchi

**Affiliations:** 1Department of Public Health, Jundiaí Medical School, Jundiaí 13202-550, SP, Brazil; carlatenani@hotmail.com (C.F.T.); adrianobarbosa@g.fmj.br (A.P.B.); 2Department of Health Sciences and Child Dentistry, Faculty of Odontology of Piracicaba, University of Campinas, Piracicaba 13414-903, SP, Brazil; carolina.matteussi@gmail.com; 3Department of Epidemiology, Faculty of Public Health, University of São Paulo, São Paulo 01246-904, SP, Brazil; mdrddola@usp.br; 4Department of Surgery, Jundiaí Medical School, Jundiaí 13202-550, SP, Brazil; evaldomarchi@g.fmj.br

**Keywords:** epidemiology, public health, SARS-CoV-2, regression analysis, mortality

## Abstract

The aims of this study were to analyze patient survival, identify the prognostic factors for patients with COVID-19 deaths considering the length of hospital stay, and evaluate the spatial distribution of these deaths in the city of Jundiaí, São Paulo, Brazil. We examined prognostic variables and survival rates of COVID-19 patients hospitalized at a reference hospital in Jundiaí, Brazil. A retrospective cohort of hospitalized cases from April to July of 2020 was included. Descriptive analysis, Kaplan–Meier curves, univariate and multivariate Cox regression, and binary logistic regression models were used. Among the 902 reported and confirmed cases, there were 311 deaths (34.5%). The median survival was 27 days, and the mean for those discharged was 46 days. Regardless of the length of hospital stay, desaturation, immunosuppression, age over 60, kidney disease, hypertension, lung disease, and hypertension were found to be independent predictors of death in both Cox and logistic regression models.

## 1. Background

With the emergence of a new type of coronavirus—SARS-CoV-2—and the World Health Organization declaration of the COVID-19 pandemic in March 2020, efforts have been made worldwide to identify the clinical and epidemiological characteristics of the infection, as well as the best measures to deal with its impacts. Within this scenario, Brazil was hardly hit, with the infection causing a large number of deaths that culminated in a lethality rate of 4.9% in June 2020 [1]. Currently—after vaccination in September 2022—the lethality rate of COVID-19 in Brazil is 2%; however, the number of deaths has reached 685,0022 [2], and Brazil is therefore one of the countries mostly affected by this disease.

Although COVID-19 initially causes mild infection [3], it is known that some patients are at higher risk of progression to severe acute respiratory syndrome (SARS), with some cases requiring hospitalization [4]. These hospitalizations and progression to death vary according to individual health conditions, the place of hospitalization, factors related to health service access, and therapeutic resources, as well as the need for ventilatory support [5]. A study conducted in the state of Piauí identified a high lethality rate among hospitalizations due to COVID-19, especially in the interior of the state. Analysis of the epidemiologic profile of these patients showed a predominance of elderly, male, and black patients with one or two comorbidities who required ventilatory support [5].

Other publications corroborated these findings, demonstrating an association between the worsening of COVID-19 disease and death with advanced age, male gender, sociodemographic factors, and the presence of comorbidities [6,7]. A survival study conducted in New York identified an association between death due to COVID-19 and the presence of higher serum interleukin concentrations [7], and, after one month, 39% of the participants had developed complications from the infection and had died [7].

The availability of medical services and beds, particularly those of intensive care with ventilatory support, was one of the greatest challenges of the pandemic facing the world [4]. In view of the above, it is necessary to estimate the length of hospitalization due to COVID-19 and also the risk factors related to the greatest risk of death for health planning and adequate care, and few studies in Brazil provide this evidence.

Hence, it is important to address how the availability of beds and the length of hospital stay may contribute to the best prognosis for patients with COVID-19. Besides that, the georeferencing must also be highlighted. In this way, the aims of this study were to analyze patient survival, identify the prognostic factors for patients with COVID-19 deaths considering the length of hospital stay, using data from the Health Information Systems, and evaluate the spatial distribution of these deaths in the city of Jundiaí, São Paulo, Brazil. Our hypothesis is that logistic regression and survival analysis can identify factors associated with mortality due to COVID-19, regardless of the length of hospitalization, and estimate the length of hospitalization due to the disease, which has impacted the entire world, occupying hospital beds in such a way as to disrupt healthcare in many parts of the world.

## 2. Methods

### 2.1. Study Design

This is a retrospective cohort study that used secondary data from notified cases of SARS hospitalized in Jundiaí, São Paulo, Brazil.

### 2.2. Study Location

The city of Jundiaí is located 57 km from the state capital, São Paulo. It is composed of 74 neighborhoods and divided into four health regions. The estimated population was 409,431 inhabitants in 2021 (a demographic density of 949.51 inhabitants/km^2^) [8]. The 2010 Human Development Index of the municipality was 0.822, a value considered to be very high compared to other Brazilian municipalities. The schooling rate for 6–14 year olds is 99.7%. According to data from Jundiaí City Hall, until 18 August 2022, there were 93,184 cases of COVID-19 and 1815 deaths. The lethality rate was 1.95% [9]. The hospital network of Jundiaí is composed of 8 hospitals, including 3 public hospitals and 5 private hospitals.

### 2.3. Population

We analyzed all notified and confirmed SARS (RT-PCR) cases that developed the first symptoms between 1 April and 31 July 2020. Only hospitalized patients, residing in the city of Jundiaí, with a diagnosis confirmed by molecular testing (RT-PCR), were included in the study. The data were collected from the epidemiological surveillance system, which rigorously follows up on patients notified and confirmed with COVID-19 (RT-PCR) from hospitalization to discharge and/or death. The database from the epidemiological surveillance system of the municipality feeds the health information systems after all confirmations. These data are reliable in regard to the diagnosis and reason for hospitalization, as well as the outcome (death or discharge).

### 2.4. Study Variables

The time of death was considered the time between the date of the first symptoms and the death or hospital discharge (censured). The prognostic factors for sociodemographic profile were (age ≤ 59 years or ≥ 60 years), sex (female or male), skin color (white or black/brown/yellow), educational level (incomplete elementary school, complete elementary school, or complete high school); presence of signs and symptoms (yes or no): fever, cough, sore throat, dyspnea, respiratory discomfort, desaturation, diarrhea, and vomiting; presence of risk factors (yes or no): heart disease, hematological diseases, liver disease, asthma, diabetes mellitus, neurological diseases, lung diseases, immunological diseases, kidney disease, arterial hypertension, and obesity. The type of care was analyzed as well as admission to intensive care unit (ICU) (yes or no) and use of ventilatory support (invasive, non-invasive, or did not use).

### 2.5. Data Analysis

The descriptive statistics (absolute and relative frequencies, means, and standard deviation—SD) were calculated. Two multiple analyses were performed. The first one was the survival analysis, considering the Kaplan–Meier method using the log-rank test and Cox regression models. The stepwise forward method was used for entry into the multiple Cox regression model. The assumption of proportionality was tested by Schoenfeld residuals. All analyses were performed using SPSS 20.0. The measure of association obtained was the hazard ratio (HR).

In the second analysis, it was not considered the time for death, using the chi-squared test or Fisher’s exact test, and the variables showing *p* < 0.20 were entered into the logistic regression model. The final model was adjusted, adopting a 95% confidence interval and a level of significance of 5%. The measure of association obtained was the odds ratio (OR).

For the analysis of spatial distribution, confirmed deaths due to SARS per neighborhood of residence were first counted. The population density of each neighborhood was then obtained according to the census carried out in 2010 [10] and the prevalence rate was calculated based on the number of confirmed cases divided by the density of each neighborhood. Finally, the number was multiplied by 100 to report the results in percentages. The spatial distribution was analyzed using the free QGIS 3.16.16 software. The cartographic bases showing the limits of the city of Jundiaí and neighborhood division were obtained from the Brazilian Institute of Geography and Statistics (IBGE) and the Department of Social Surveillance (SEMADS), respectively [10].

### 2.6. Ethical Aspects

The study was approved by the Research Ethics Committee of the Faculty of Medicine of Jundiaí (#4.040.674), followed by approval by the Municipal Health Secretariat. The study was conducted in accordance with the guidelines of Resolution 466/2012. Informed consent was not obtained since this study used secondary data that would not permit identification of the participants.

## 3. Results

During the follow-up period, 902 hospitalizations due to SARS were reported, and 99.8% of these cases had a positive PCR. Among all cases analyzed, 31 (3.4%) were hospitalized at the end of the study, 560 (62.1%) were discharged, and 311 (34.5%) had died.

Most patients were men (57.6%), older adults (52.8%), and white (73.8%). The main signs and symptoms were fever (68.7%), cough (84.3%), dyspnea (82.7%), respiratory discomfort (68.5%), and desaturation (85.1%).

Table 1 shows the factors associated with death. The significant factors were age ≥60 years, presence of signs and symptoms (fever, cough, sore throat, and desaturation), presence of comorbidities (problems in heart, liver, neurological, lung, immunological, and kidney disease, asthma, diabetes mellitus, and arterial hypertension), and the type of care (ICU admission and use of ventilatory support).

The cumulative probability of survival of hospitalized patients was 77% after 15 days of hospitalization and 15.8% after 133 days. The median survival was 27 days, and the mean length of hospital stay was 46 days (Figure 1).

Cox regression showed that age over 60 years (HR = 2.46), desaturation (HR = 1.62), immunosuppression (HR = 1.83), kidney disease (HR = 2.02), hypertension (HR = 1.65), and lung disease (HR = 1.61) were risk factors for death, regardless of length of hospital stay (Table 2).

In the adjusted logistic regression model, the risk of progression of SARS to death was associated with age ≥ 60 years (OR = 4.18), desaturation (OR = 2.29), and presence of neurological (OR = 3.55), lung (OR = 3.24), immunological (OR = 2.84), and kidney disease (OR = 6.08) (Table 3).

Analysis of spatial distribution revealed cases of death due to COVID-19 in 45 neighborhoods; however, the prevalence was higher in Vila Municipal (0.27%), Ivoturucaia (0.22%), Champirra (0.21%), Distrito Industrial, and Hortolândia (both with 0.16%), as illustrated by the darker color in the map (Figure 2).

## 4. Discussion

The results revealed that hospital death due to COVID-19 in Jundiaí/SP was associated with older age and the presence of risk factors such as chronic diseases, immunosuppression, and signs/symptoms such as desaturation. The survival probability was 50% at about 27 days, dropping to 15% at the end of the study (133 days). Cox regression allowed for the identification of hypertension in addition to the variables that were found to be associated with logistic regression. Regarding spatial distribution, although present throughout the area studied, the prevalence of cases that progressed to death was higher in the Vila Municipal neighborhood. This study brings relevant results and will provide epidemiological data and data on space–time dynamics that will serve as a tool to identify particularities of the territory and consequently guide actions to support health decisions targeting local needs [11].

The epidemiology profile of hospitalized patients is that most were men, older than 60 years old, with chronic disease. These data are consistent with the national and international literature found, which may indicate a worse prognosis for these related conditions [1,3,12].

The association between SARS-related death due to COVID-19 and demographic characteristics such as age over 60 years and sex (only in univariate analysis) identified in this study corroborates national and international findings [5,13,14,15,16,17]. A study conducted in the state of Piauí [18] found that 57.1% of hospitalizations due to COVID-19 were 60 years of age or older. The lethality rate in this age group was 45%. Another study conducted in the state of Rio Grande do Norte [17] also reported a predominance of deaths among men (55.4%) and patients aged 60 to 79 years (43.2%) and 80 years or older (28.6%), as well as an age-related increase in the lethality rate. In the present study, age was a risk factor in both analyses. Older age was identified as the main risk factor for developing the most severe form of COVID-19 and for death, which can be explained by the fact that this age group has more comorbidities (which are also risk factors) as a result of the weakening of the immune system that occurs with aging and increased production of inflammatory cytokines [19,20]. In the study by Bucholc et al. [20], considering patients without a history of comorbidities, those who died in the hospital were significantly older than those who were discharged, showing that age is a prognostic factor even among patients without chronic comorbidities.

Another factor increasing the risk of death identified in the municipality was chronic neurological diseases, in agreement with the study by Berenguer et al. (2020) [21]. Age over 65 years and the presence of cardiovascular or cerebrovascular diseases have also been reported as predictors of death due to COVID-19 [22]. It is important to note that neurological diseases have been identified as a condition that increases the risk of death and as a post-COVID-19 complication, with one study showing an increased risk of ischemic and cryptogenic stroke and increased mortality risk [23]. Other COVID-19 complications have also been identified, which are mainly the result of the inflammatory process triggered by SARS-CoV-2 [24]. Another study indicated dementia as a risk factor for in-hospital mortality, reinforcing the role of chronic neuropathy as a predictor of poor prognosis in patients with COVID-19 [20]. In that study, analysis by disease cluster showed higher mortality in the group of patients with concomitant mental diseases and neurological or cardiovascular diseases.

The survival evidenced in this study was different from that found by Cumming et al. [7]. The authors observed a mean length of hospital stay of 19 days in critically ill patients with COVID-19, and 39% of the hospitalized patients had died at the end of the study. This difference in length of stay compared to the literature may be due to the fact that the American study was conducted at the beginning of the pandemic, when little was known about the management of the disease. An estimation of the length of hospitalization due to the disease is very important because this problem has impacted the entire world, causing a health crisis in many parts of the world.

In this study, Cox regression analysis demonstrated an association between hypertension and death, agreeing with the study by Berenguer et al. (2020) [21]. The inflammatory process triggered systemically by SARS-CoV-2 may explain the increased risk of death and the development of the severe form of the disease in patients with comorbidities such as hypertension [19].

Because of the inflammatory process involved in the pathophysiology of SARS infection, the use of immunosuppressants was debated at the beginning of the pandemic; it was then quickly concluded that, for example, hydroxychloroquine did not alter patient mortality [25]. On the other hand, the RECOVERY study [26] demonstrated clear benefits of the use of dexamethasone, especially in patients undergoing mechanical ventilation. The presence of other comorbidities, such as immunosuppression and chronic diseases, was considered a risk factor for COVID-19 and was associated with the progression of the condition to death. Similar results have been reported in the international [16,19,21,27] and national literature [5,15]. Furthermore, immunosuppressed patients in the present study had an increased risk of death, as also reported by Gao et al. (2020) [19], who found immunosuppression to be associated with the most severe form of the disease and with the risk of mortality, although studies did not observe a greater risk of contracting the infection [19].

Angiotensin-converting enzyme 2 (ACE2) serves as the functional host receptor for SARS-CoV and SARS-CoV-2 [28]. An imbalance between the two main pathways of the renin–angiotensin–aldosterone system (down-regulated ACE2/angiotensin-(1–7) and up-regulated ACE/angiotensin II) and the cytokine storm triggering an inflammatory process may explain the increased risk of severe disease in COVID-19 patients with comorbidities and advanced age [16], as observed in the present study. Cumming et al. (2020) [7] also observed a higher risk of death in hospitalized patients with high levels of interleukin 6 (which is a proinflammatory mediator). At the beginning of the pandemic, researchers believed that the use of ACE inhibitors and angiotensin-receptor blockers (ARBs), widely used as first-line antihypertensive treatment, could be associated with a higher risk of hospitalization or death. Several studies have shown that ACE2 expression is low in the lower respiratory tract. ACE inhibitors and ARBs were found to protect against the deleterious proinflammatory effects mediated by angiotensin II. Furthermore, treatment with ACE inhibitors was associated with an intrinsic antiviral response. In the study of Gallo G et al. (2022), the use of ACE inhibitors/ARBs or their combination with other antihypertensive agents was not significantly associated with COVID-19 or a more severe course of the disease [29].

The Cox regression shows a hazard ratio of death for patients with kidney disease of 2.02, while a study conducted in Spain [21] found a hazard ratio of 1.55, with both values being significant. A higher risk of death has been reported for patients with chronic kidney disease. The authors identified a high prevalence of comorbidities such as hypertension, cardiovascular diseases, and diabetes mellitus in patients with chronic kidney disease, which can contribute to the worsening of the condition [19].

The same line of reasoning can be applied to the observation that lung diseases do not predispose to the disease; however, patients with this condition are at greater risk of complications and death, as observed in the present study [18,21]. Guan et al. [30] showed that chronic obstructive pulmonary disease (COPD) was a risk factor for ICU admission, invasive ventilation, and death in a Chinese population after adjusting for age and smoking. The increased risk of severe disease and adverse outcome in patients with COVID-19 and co-existing COPD can be attributed to reduced lung reserve, increased ACE2 expression in the bronchial epithelium, chronic lung inflammation, chronic hypoxemia, destruction of the lung parenchyma, expiratory flow limitation, acute exacerbation by viral infection, mucus hypersecretion, and pulmonary hypertension [19].

All the cited factors can contribute to oxygen desaturation and an increased risk of serious outcomes in patients with COVID-19, as was also observed in this study and in Spain [21]. According to a study conducted in 2020 on patients admitted to hospitals in the United Kingdom [31], the lethality rate was higher than 26% in patients with low oxygen saturation, which was a risk factor for progression to death. It is noteworthy that SARS due to COVID-19 requires attention; in the case of patients in the present study, hospital care was necessary. Worsening of the condition in respiratory infections can lead to hypoxia, which requires ventilatory support; the latter is a factor of poor prognosis and prolonged ventilation that may lead to death [31]. Adopting preventive measures and promoting population health are therefore important to reduce and control risk factors.

The global pandemic has had severe social and health impacts worldwide. An important discussion was about the environmental impact of the pandemic, which led to changes in clinical and surgical settings. It is known that healthcare contributes to 4.9 percent of the world’s carbon emissions [32]. During the pandemic, elective surgeries were cancelled. What could contribute to carbon emissions once the top three ranked interventions to reduce the environmental impact of operating theaters were introducing reusable surgical devices, reducing the use of consumables, and reducing the use of general anesthesia [33]. However, due to heavily crowded hospitals during the pandemic, this may not have happened, and the need for procedures such as ventilatory support, both invasive and non-invasive, may have further increased the consumption of inputs and the need for anesthetics, among other patient care actions that further impacted the environment. Another impact was the increasing use of hand soap and other hygiene products globally [34] Thinking about the hospitalization, an important point to be considered was the negative effects of COVID-19 on increasing medical waste [35].

Limitations of the present study include the use of secondary data from the epidemiological surveillance system database of SARS notification forms for hospitalized cases, as well as its retrospective design. As all studies were carried out with secondary data, there are limitations in regards to detailed information such as socio-economic status, access to healthcare facilities, and detailed patient histories (e.g., smoking status, obesity) that could not be thoroughly explored.

The retrospective nature limits the ability to establish causality, and the study covers a four-month period, which may not capture the full progression of the pandemic. Despite the difficulty of temporal establishment in a retrospective study, in this case, the date of the notification of the disease (initial symptoms and RT-PCR) and the date of discharge or death were reliable information with a confident temporal relationship.

Despite these limitations, the use of multiple logistic regression analysis, complemented by survival analysis, Cox regression, and spatial analysis, provided robust results that are of great importance for the scientific community and health managers. We recognize the possible gaps in all analyses carried out. However, the use of more than one method improved the quality of the results. The study covers various demographic, clinical, and epidemiological factors, providing a detailed understanding of COVID-19 mortality and length of hospitalization. In the present study, the results must be interpreted carefully, considering the internal validity because it is not a multicenter study.

In addition to the limitations, the data comprise a period prior to vaccination and may be attributed to individuals who did not receive the complete vaccination schedule. In Brazil, 19.5% of the eligible population did not take the second dose, and 52.3% did not take the booster. Globally, only 63% of the population is vaccinated, while in low-income countries, only 22.6% have received a vaccine dose [13]. Following vaccination, further clinical risk factors for severe COVID-19 outcomes have been identified, including Down syndrome, kidney transplant, sickle cell disease, living in a nursing home, chemotherapy, recent bone marrow transplant, history of solid organ transplants, HIV/AIDS, dementia, Parkinson’s disease, neurological conditions, and liver cirrhosis [36]. Within this context, we observed that some conditions, such as immunosuppression and neurological diseases, remain a risk factor even after vaccination.

Regarding the spatial distribution of the prevalence of deaths, according to data from the Brazilian Institute of Geography and Statistics (IBGE) census (2010) [10], the Vila Municipal and Ivoturucaia neighborhoods—with the highest prevalence of deaths—had a larger number of older adults in 2010, as well as a higher per capita income (USD 5.74 and USD 2.77, respectively). A seroepidemiological survey conducted in Jundiaí demonstrated a higher percentage of positive COVID-19 cases at the periphery of the municipality and found a spatial correlation between positive cases and neighborhoods with higher per capita income [37]. The higher prevalence of deaths in these neighborhoods may have been due to the age transition of their population. According to Jundiaí City Hall, the municipality has shown an increase in the aging rate of its population—100.96% in 2021 [9], highlighting the need for adapting health services to meet the demands of this population and its health conditions. Spatial analysis was very important for the health managers, especially in the pandemic period.

Future studies should consider prospective design with a longer follow-up period and the possibility of including other variables of interest as socioeconomic variables. It is important to include those who have been fully vaccinated, those who have not been vaccinated, and how many doses.

Therefore, effective public health responses, as well as greater scientific understanding of SARS-CoV-2, are necessary to control and mitigate SARS-related deaths due to COVID-19. Given this scenario, it is important to think about actions that will alert health authorities to preventive measures and promote population health, highlighting the benefits of vaccination and the development of risk-oriented protocols considering age group, comorbidities, and vulnerabilities [14,19]. Exploring methods to analyze a database could improve health information, and it is necessary to emphasize the importance of science for decision making, which was a problem observed around the world during the pandemic. Finally, this study, which addressed the length of hospitalization and the risk of mortality due to COVID-19, should be taken into consideration when planning public health. May the pandemic be a learning experience to improve future actions in situations of high impact with a high number of hospitalizations and mortality, as we lived.

## 5. Conclusions

The median survival was 27 days in the sample of patients hospitalized during the study period (133 days). There was an association between SARS-related death due to COVID-19 and older age, low saturation, and the presence of comorbidities such as hypertension, immunosuppression, lung disease, and kidney disease. The use of two types of regression analysis permitted for the identification of an additional clinical variable as a risk factor for COVID-19 death. Survival analysis and death-related factors can contribute to the planning of actions and the reorganization of health services in order to allocate resources and meet the needs of the population at risk. Epidemiological and scientific information should be the basis for decision making in public health and should guide public health policies. This study highlights the importance of analyzing risk factors and the average length of hospital stay and testing statistical methods to subsidize public health actions.

## Figures and Tables

**Figure 1 ijerph-21-01211-f001:**
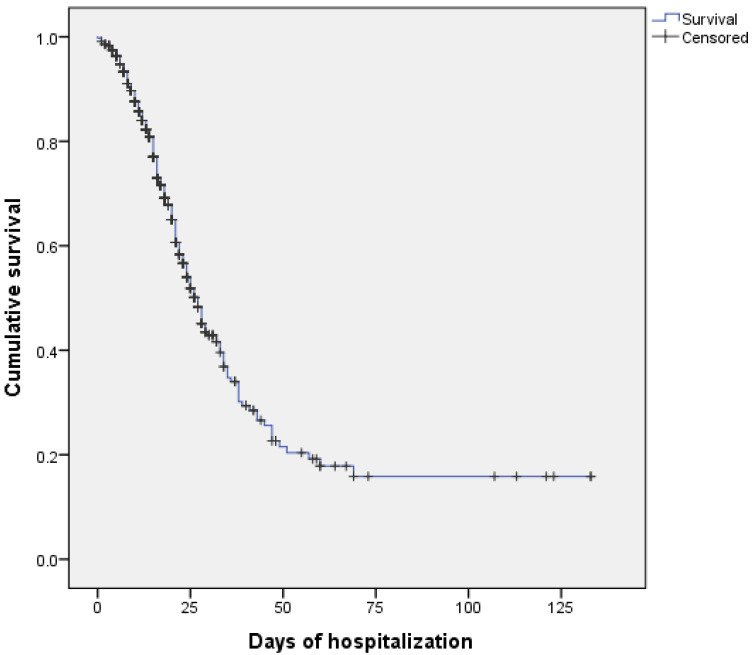
Kaplan–Meier cumulative survival curve of hospitalized COVID-19 patients, Jundiaí-SP, 2020.

**Figure 2 ijerph-21-01211-f002:**
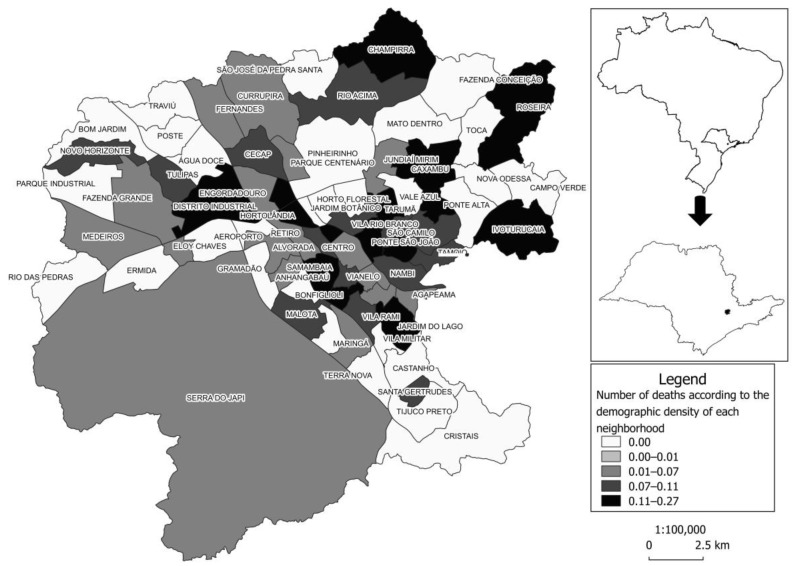
Prevalence of cases of death due to COVID-19 (*n* = 311) according to demographic density in each neighborhood, Jundiaí-SP, 2020. Note: The point on the map indicates the location in the municipality in the State of São Paulo, Brazil.

**Table 1 ijerph-21-01211-t001:** Distribution of the profile of cases of severe acute respiratory syndrome and associated factors according to outcome (discharge or death) (*n* = 871), Jundiaí-SP, 2020.

Variable		Total ^a^	Discharge	Death	*p*-Value
		*n* (%)	*n* (%)	*n* (%)	
Epidemiological profile					
Age	≤59 years	426 (47.2)	331 (59.1)	71 (22.8)	<0.001 ^a^
	≥60 years	476 (52.8)	229 (40.9)	240 (77.2)	
Sex	Female	381 (42.4)	238 (42.7)	133 (42.9)	0.943 ^a^
	Male	518 (57.6)	320 (57.3)	177 (57.1)	
Skin color	White	571 (73.8)	335 (72.7)	216 (74.2)	0.638
	Black/brown/yellow	203 (26.2)	126 (27.3)	75 (25.8)	
Educational level	Incomplete elementary	9 (19.1)	6 (18.8)	2 (40.0)	0.536 ^b^
	Complete elementary	1 (2.1)	1 (3.1)	0 (0.0)	
	Complete high school	37 (78.7)	25 (78.1)	3 (60.0)	
Presence of symptoms					
Fever	Yes	553 (68.7)	360 (70.9)	170 (63.4)	0.034 ^a^
	No	252 (31.3)	148 (29.1)	98 (36.6)	
Cough	Yes	708 (84.3)	458 (86.9)	225 (78.9)	0.003 ^a^
	No	132 (15.7)	69 (13.1)	60 (21.1)	
Sore throat	Yes	115 (17.2)	84 (19.5)	22 (10.3)	0.003 ^a^
	No	552 (82.8)	346 (80.5)	192 (89.7)	
Dyspnea	Yes	691 (82.7)	424 (81.2)	245 (86.0)	0.087 ^a^
	No	145 (17.3)	98 (18.8)	40 (14.0)	
Respiratory discomfort	Yes	516 (68.5)	313 (66.7)	185 (71.4)	0.192 ^a^
	No	237 (31.5)	156 (33.3)	74 (28.6)	
Desaturation	Yes	684 (85.1)	408 (82.3)	256 (91.4)	<0.001 ^a^
	No	120 (14.9)	88 (17.7)	24 (8.6)	
Diarrhea	Yes	145 (21.3)	94 (21.8)	48 (21.2)	0.878 ^a^
	No	535 (78.7)	338 (78.2)	178 (78.8)	
Vomiting	Yes	45 (6.8)	32 (7.6)	10 (4.6)	0.145 ^a^
	No	613 (93.2)	387 (92.4)	207 (95.4)	
Presence of risk factors					
Heart disease	Yes	346 (38.4)	189 (33.8)	151 (48.6)	<0.001 ^a^
	No	555 (61.6)	370 (66.2)	160 (51.4)	
Hematological disease	Yes	6 (0.7)	3 (0.5)	3 (1.0)	0.672 ^b^
	No	896 (99.3)	557 (99.5)	308 (99.0)	
Liver disease	Yes	5 (0.6)	0 (0.0)	5 (1.6)	0.006 ^b^
	No	897 (99.4)	560 (100.0)	306 (98.4)	
Asthma	Yes	22 (2.4)	19 (3.4)	3 (1.0)	<0.029 ^a^
	No	880 (97.6)	541 (96.9)	308 (99.0)	
Diabetes mellitus	Yes	249 (27.6)	131 (23.4)	110 (35.4)	<0.001 ^a^
	No	653 (72.4)	429 (76.6)	201 (64.6)	
Neurological disease	Yes	53 (5.9)	17 (3.0)	34 (10.9)	<0.001 ^a^
	No	849 (94.1)	543 (66.2)	277 (89.1)	
Lung disease	Yes	58 (6.4)	20 (3.6)	38 (12.2)	<0.001 ^a^
	No	844 (93.6)	540 (96.4)	273 (87.8)	
Immunological disease	Yes	41 (4.5)	10 (1.8)	29 (9.3)	<0.001 ^a^
	No	861 (95.5)	550 (98.2)	282 (90.7)	
Kidney disease	Yes	37 (4.1)	550 (98.2)	282 (90.7)	<0.001 ^a^
	No	865 (95.9)	10 (1.8)	25 (8.0)	
Arterial hypertension	Yes	103 (11.4)	550 (98.2)	286 (92.0)	0.014 ^a^
	No	799 (88.6)	55 (9.8)	48 (15.4)	
Obesity	Yes	98 (10.9)	58 (10.4)	35 (11.3)	0.681 ^b^
	No	804 (89.1)	502 (89.6)	276 (88.7)	
Type of care					
ICU admission	Yes	212 (24.6)	100 (18.3)	102 (35.1)	<0.001 ^a^
	No	650 (75.4)	445 (81.7)	189 (64.9)	
Use of ventilatory support	Invasive	85 (10.2)	19 (3.7)	64 (22.1)	<0.001 ^a^
	Non-invasive	524 (62.9)	320 (62.0)	184 (63.7)	
	Did not use	224 (26.9)	177 (34.3)	41 (14.2)	

*n* is not 871 for some variables because of lost/missing data. ^a^ Chi-square test. ^b^ Fisher.

**Table 2 ijerph-21-01211-t002:** Cox regression for death as outcome in patients hospitalized due to COVID-19, Jundiaí-SP, 2020.

Block	Variable		Unadjusted HR (95% CI)	*p*-Value	Adjusted HR (95% CI)	*p*-Value
Sociodemographic	Age	≥60 years	2.78 (2.17–3.57)	<0.001	2.46 (1.87–3.22)	<0.001
		≤59 years	1.00		1.00	
Symptoms	Fever	Yes	0.82 (0.64–1.05)	0.113		
		No	1.00			
	Cough	Yes	0.73 (0.55–0.97)	0.03		
		No	1.00			
	Sore throat	Yes	0.54 (0.34–0.83)	0.006		
		No	1.00			
	Dyspnea	Yes	1.24 (0.89–1.74)	0.205		
		No	1.00			
	Respiratory discomfort	Yes	1.21 (0.92–1.59)	0.165		
		No	1.00			
	Desaturation	Yes	1.91 (1.26–2.91)	0.003	1.62 (1.12–2.30)	0.01
		No	1.00		1.00	
	Vomiting	Yes	0.69 (0.37–1.31)	0.260		
		No	1.00			
Presence of risk factors	Heart disease	Yes	1.41 (1.13–1.77)	0.002		
		No	1.00			
	Asthma	Yes	0.42 (0.14–1.31)	0.135		
		No	1.00			
	Diabetes mellitus	Yes	1.38 (1.01–1.75)	0.006		
		No	1.00			
	Neurological disease	Yes	1.76 (1.23–2.52)	0.002		
		No	1			
	Lung disease	Yes	2.05 (1.46–2.89)	<0.001	1.61 (1.21–2.30)	0.01
		No	1.00		1.00	
	Immunological disease	Yes	2.28 (1.56–3.35)	<0.001	1.83 (1.22–1.73)	0.003
		No	1.00		1.00	
	Kidney disease	Yes	2.25 (1.49–3.38)	<0.001	2.02 (1.29–3.15)	0.002
		No	1.00		1.00	
	Arterial hypertension	Yes	1.76 (1.29–2.40)	<0.001	1.65 (1.19–2.27)	0.002
		No	1.00		1.00	
	Obesity	Yes	0.90 (0.63–1.27)	0.537		
		No	1.00			
Type of care	ICU admission	Yes	1.28 (1.01–1.63)	0.046		
		No	1.00			
		Yes	2.45 (1.84–3.26)	<0.001		
		No	1.00			

**Table 3 ijerph-21-01211-t003:** Logistic regression for death as outcome in patients hospitalized due to COVID-19, Jundiaí-SP, 2020.

Block	Variable		Unadjusted OR(95% CI)	*p*-Value	Adjusted OR(95% CI)	*p*-Value
Sociodemographic	Age	≥60 years	4.88 (3.57–6.68)	<0.001	4.18 (2.80–6.25)	<0.001
		≤59 years			1.00	
Symptoms	Fever	Yes	0.71 (0.52–0.98)	0.035		
		No	1.00			
	Cough	Yes	0.56 (0.39–0.83)	0.035	0.64 (0.40–1.03)	0.064
		No	1.00		1.00	
	Sore throat	Yes	0.47 (0.29–0.78)	0.035		
		No	1.00			
	Dyspnea	Yes	1.41 (0.95–2.11)	0.088		
		No	1.00			
	Respiratory discomfort	Yes	1.25 (0.89–1.73)	0.193		
		No	1.00			
	Desaturation	Yes	2.30 (1.43–3.71)	0.001	2.29 (1.29–4.07)	0.005
		No	1.00		1.00	
	Vomiting	Yes	0.58 (0.28–1.21)	0.149	0.41 (0.16–1.03)	0.058
		No	1.00		1.00	
Presence of risk factors	Heart disease	Yes	1.85 (1.39–2.45)	<0.001		
		No	1.00			
	Asthma	Yes	0.28 (0.81–0.94)	0.040		
		No	1.00			
	Diabetes mellitus	Yes	1.79 (1.32–2.43)	<0.001		
		No	1.00			
	Neurological disease	Yes	3.92 (2.15–7.14)	<0.001	3.55 (1.47–8.59)	0.005
		No	1.00		1.00	
	Lung disease	Yes	3.76 (2.15–6.58)	<0.001	3.24 (1.54–6.80)	0.002
		No	1.00		1.00	
	Immunological disease	Yes	5.66 (2.72–11.77)	<0.001	2.84 (1.14–7.10)	0.025
		No	1.00		1.00	
	Kidney disease	Yes	4.81 (2.28–10.15)	<0.001	6.08 (1.98–18.63)	0.002
		No	1.00		1.00	
	Arterial hypertension	Yes	1.67 (1.11–2.54)	0.015		
		No	1.00			
	Obesity	Yes	1.06 (0.69–1.65)	0.785		
		No	1.00			
Type of care	ICU admission	Yes	2.40 (1.74–3.32)	<0.001		
		No	1.00			
		Yes	5.63 (3.33–9.52)	<0.001		
		No	1.00			

## Data Availability

The original data are openly available in Open Science Framework (OSF) repository via the link: https://osf.io/amsxn/?view_only=46449a9acdfc44219551b5cfd56d8a62, accessed on 20 August 2024.

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
