# Peer review of "COVID-19 Mortality among Hospitalized Patients: Survival, Associated Factors, and Spatial Distribution in a City in São Paulo, Brazil, 2020"

_ijerph, 2024, doi:10.3390/ijerph21091211_

Round 1

Reviewer 1 Report (Previous Reviewer 1)

Comments and Suggestions for Authors

Thanks for the authors for considering reviewers' comments and recommendations. In my opinion, now the paper can be accepted.

Author Response

  • Comments 1: Thanks for the authors for considering reviewers' comments and recommendations. In my opinion, now the paper can be accepted.

Reply: Thank you very much for your comments that improved the manuscript.

Reviewer 2 Report (Previous Reviewer 3)

Comments and Suggestions for Authors

The authors resubmit the original manuscript without modification in light of comments from the first revision. The response to the overarching comments from the first submission engage in ipse dixit reasoning to continue to engage in statistical analyses that are invalid. The addition of a few sentences to the limitations section that state that there are limitations is not compelling. 

Comments on the Quality of English Language

Errors did not interrupt overall flow.

Author Response

  • The authors resubmit the original manuscript without modification in light of comments from the first revision. The response to the overarching comments from the first submission engage in ipse dixit reasoning to continue to engage in statistical analyses that are invalid. The addition of a few sentences to the limitations section that state that there are limitations is not compelling. 

Reply: We, the authors, sent a letter responding point by point to the reviewers' recommendations and questions including statistical analysis and its limitations. We have worked very hard to attend all of the reviewers comments. We agree that it would be very important to have other data and a longer follow-up period, but in this study it is not possible because the information was collected through the Epidemiological Surveillance notification forms, and this is a limitation for all studies that using secondary data. Because of this we added the limitations and suggestions for future studies. 

With my respect, to say that the logistic regression and survival analyses are invalid, I believe, is a very strong statement, since they are considered robust analyses and were pointed out as a strenght of the study.

Reviewer 3 Report (New Reviewer)

Comments and Suggestions for Authors

The paper performs an exercise to model deaths due to COVID-19 in Jundiai, Sao Paulo, using Cox and Logistic regressions. The results show that there are factors that help explain deaths, such as comorbidities and socio-demographic characteristics.  

The methods used are sound and standard in the literature.

An ethics committee has approved the research. As the study employs secondary data, information consent is not required.

Main points:

  1. I wonder whether the data would be available in a repository so other researchers can use it to test different hypotheses. The potential for citing the current paper increases if the data is available.
  2. The conclusion section is concise. Discussing how the results may translate into effective public policies to fight pandemics would be very important or at the very least explaining what have we learned after reading the paper on how to deal with similar pandemics if they occurr again.
  3. An additional paragraph explaining the limitations of the paper and what is left for further research would add value to the paper.

Overall, this is a very interesting paper with important results. 

Author Response

  • The paper performs an exercise to model deaths due to COVID-19 in Jundiai, Sao Paulo, using Cox and Logistic regressions. The results show that there are factors that help explain deaths, such as comorbidities and socio-demographic characteristics.  

The methods used are sound and standard in the literature.

An ethics committee has approved the research. As the study employs secondary data, information consent is not required.

 Reply: Thank you very much. The modifications carried out were made highlighted in blue.

Main points:

Comment 1: I wonder whether the data would be available in a repository so other researchers can use it to test different hypotheses. The potential for citing the current paper increases if the data is available.

Reply: The data is available. Data accessible in Open Science Framework (OSF) repository via the link: https://osf.io/amsxn/?view_only= 46449a9acdfc44219551b5cfd56d8a62.

Comment 2: The conclusion section is concise. Discussing how the results may translate into effective public policies to fight pandemics would be very important or at the very least explaining what have we learned after reading the paper on how to deal with similar pandemics if they occurr again.

Reply: We agree about the relevance of this topic in conclusion as well. We added this discussion at the end of the manuscript.

Comment 3: An additional paragraph explaining the limitations of the paper and what is left for further research would add value to the paper.

Reply: We added one paragraph clarifying the limitations and the recommendation of further studies.

Comment 4: Overall, this is a very interesting paper with important results. 

Reply: Thank you much, the reviewer comments have improved the manuscript.

This manuscript is a resubmission of an earlier submission. The following is a list of the peer review reports and author responses from that submission.

Round 1

Reviewer 1 Report

Comments and Suggestions for Authors

The paper comprehensively analyses the factors associated with COVID-19 mortality among hospitalized patients in Jundiaí, São Paulo, Brazil. It utilizes a retrospective cohort study design, analyzing data from April to July 2020. The study employs descriptive analysis, Kaplan-Meier curves, Cox regression, and logistic regression to identify prognostic factors and spatial distribution of deaths.

Strengths:

1. The use of multiple analytical methods (Kaplan-Meier, Cox regression, logistic regression) strengthens the validity of the findings.

2. The study covers various demographic, clinical, and epidemiological factors, providing a detailed understanding of COVID-19 mortality.

3. Including spatial distribution offers valuable insights into how geographic factors influence COVID-19 outcomes.

4. Tables and figures are well-organized, making the data easily interpretable.

Drawbacks:

1. The retrospective nature limits the ability to establish causality and is prone to biases inherent in secondary data.

2. The study only covers a four-month period, which may not capture the full progression of the pandemic and its changing dynamics over time.

3. The data predate the widespread availability of COVID-19 vaccines, which limits the findings' applicability to current contexts.

4. Important factors such as socio-economic status, access to healthcare facilities, and detailed patient histories (e.g., smoking status, obesity) are not thoroughly explored.

5. Although 902 cases are analyzed, the sample size may still be insufficient to generalize findings to larger populations or different settings.

Recommendations:

1. Conduct prospective studies to confirm causality and mitigate biases associated with retrospective data.

2. Expand the study period to include different phases of the pandemic, including post-vaccination periods, to understand the long-term trends and impacts better.

3. Incorporate socio-economic data to evaluate how these factors influence COVID-19 mortality, which could provide a more holistic understanding.

4. To enhance the generalizability of the findings, increase the sample size. Multi-centre studies could be conducted to include diverse populations.

Reviewer 2 Report

Comments and Suggestions for Authors

Abstract:
- The abstract should enumerate the main conclusions and explicitly define the goals of the study. To save space, certain information regarding the study's methodology may be eliminated. The writing has to be more succinct. For instance: "We examined prognostic variables and survival rates of COVID-19 patients hospitalized to a reference hospital in Jundiaí, Brazil. A retrospective cohort of hospitalized cases from April to July of 2020 was included. Regardless of the length of hospital stay, desaturation, immunosuppression, age over 60, kidney disease, hypertension, lung disease, and hypertension were found to be independent predictors of death in both Cox and logistic regression models."

Introduction:
- The introduction gives useful background information, but it is devoid of details regarding the purpose and particular goals of the current investigation. This part would be strengthened with an additional paragraph that explicitly states the study questions, hypotheses, and aims.
- Some details concerning epidemiology in other places are not immediately relevant, thus the wording should be more targeted.

Methods:
- In general, the methods section offers sufficient information regarding the population, variables, analysis, setting, and study design. A few points require explanation:
  - It is necessary to outline the requirements for both a SARS diagnosis and hospital admission.
  - The presentation of the results and the statistical analysis techniques should happen in the same sequence. For instance, it is best to explain survival analysis techniques prior to logistic regression.
  - The techniques section should end with the ethical approval procedures.

Results:
Using tables and figures, the data are rationally presented and well-organized.
- It would be easier to understand the main findings if some summary statistics, such as odds or hazard ratios, were included in the text itself.
- Certain sections of the text might be shortened, such as the part where it states that a factor was statistically significant without needing to include the precise p-values.

Discussion:
- The discussion should expand on interpreting the key findings, comparing them to prior literature, noting study limitations, and discussing implications.
The conversation as it is now is not very deep.

- discuss how the clinical and surgical settings changed during COVID-19 and how it impacted on carbon print production. discuss and cite     doi:10.1093/bjs/znad092
- The authors ought to go into further information about how their results compare to those of other research on prognostic factors in COVID-19 individuals. Do they exhibit consistency?

Comments on the Quality of English Language

none

Reviewer 3 Report

Comments and Suggestions for Authors

The submission summarizes several statistical analyses of the risk of death post hospitalization for COVID-19 in a Brazilian city. The authors found several covariates associated with an elevated risk of death to include age, race, and comorbidities. This is an analysis of extant data that follows patients from the point of hospital admission to death, discharge from hospital, or the end of the follow up. The authors use both logistic regression and Cox proportional hazards to analyze death as the outcome. Unfortunately, both approaches are fatally flawed given the data. The logistic model does not account for censoring and assumes that those who were censored did not have the outcome of interest. Conversely, the Cox model assumes that censoring is not related to the risk of the outcome in question. Since more than one-half of the study sample were discharged from the hospital, which was the censoring point, it can be assumed that they were in a better health status at the point of discharge than a patient who remained inpatient. These individuals would have a lower risk of death, so the analysis of the outcome only in patients who remain inpatient is biased and the violation of the survival analysis assumption cannot be remediated. More minor concerns relate to the use of p-values, which is not recommended, the reliance on screening/stepwise regression (to be avoided in favor of full adjustment since the goal of the study is not to develop a predictive algorithm), and the addition of the spatial analysis which feels like an afterthought.

Comments on the Quality of English Language

Language quality generally good. There are instances of incorrect usage (e.g., 'hardly hit' should be rephrased as 'hit hard') that mildly interrupt the flow of the article.